# Cohort profile: Mothers who use substances and their children in British Columbia, Canada

**Lindsay A. Wilson[1], Noah Katsuno[2,3], Megan Kurz[2,3], Nicole Catherine[3], Shannon Joyce[2,3], Brittany Barker[3], Brittany B. Dennis[4,5], Sarah Moreheart[6], Lianping Ti[4,5], Jane A. Buxton[1], Bohdan Nosyk[2,3]***

1 School of Population and Public Health, University of British Columbia, Vancouver, British Columbia, Canada, 2 Health Economic Research Unit, Centre for Health Evaluation and Outcome Sciences, Vancouver, British Columbia, Canada, 3 Faculty of Health Sciences, Simon Fraser University, Burnaby, British Columbia, Canada, 4 Faculty of Medicine, University of British Columbia, Vancouver, British Columbia, Canada, 5 British Columbia Centre on Substance Use, Vancouver, British Columbia, Canada, 6 BC Women's Hospital + Health Centre, Vancouver, British Columbia, Canada

* bnosyk@sfu.ca

## Abstract

### Introduction

Perinatal substance use is a growing public health challenge in the province of BC. A population-based cohort was established using linked health administrative data to underpin three research studies that will evaluate: 1) the effectiveness of specialized acute care for pregnant people who use substances; 2) the comparative effectiveness of different medication regimens for the treatment of perinatal opioid use disorder (OUD); and 3) the longitudinal association between maternal substance use and child health outcomes.

### Methods

The population-based cohort includes all individuals in British Columbia (BC), Canada, who delivered an infant from April 1, 2000–March 31, 2022, and who had an indication of substance use in their health administrative records between one year prior to the first pregnancy-related healthcare contact and time of delivery. Individual-level data for mothers and children were linked across ten population-level databases, which include information on demographics, healthcare utilization, drug dispensations, incarceration in provincial prisons, maternal and newborn health outcomes, receipt of housing and income assistance, and deaths and underlying causes.

### Results

We identified 38,670 mothers with substance use disorders and their children (n = 45,823), with a median of 9 (interquartile range: 5–15) years of follow-up

**Data availability statement:** De-identified data are not available for public access, according to the British Columbia Ministry of Health and the UBC Research Ethics Board. Access to data provided by the Data Steward(s) is subject to approval, but can be requested for research projects through the Data Steward(s) or their designated service providers (See: https://healthdataplatformbc.ca/apply-data-access).

**Funding:** Funding for this cohort was obtained from the Health Canada Substance Use and Addictions Program (2223-HQ-000028; 1819-HQ-000036). The funding source was independent of the design of this study and did not have any role during its execution, analyses, interpretation of the data, writing, or decision to submit results.

**Competing interests:** The authors have declared that no competing interests exist.

available. At the time of delivery, mothers ranged from <19–45 years of age. High levels of morbidity and mortality were documented among both mothers and their children.

## Discussion

The growing population of mothers with substance use disorder represent a critical need for the province of British Columbia. This population-based retrospective cohort will be used to inform service design and otherwise advocate for client needs.

## Introduction

Substance use during pregnancy and post-partum (hereafter known as perinatal substance use [PSU]) is associated with increased risk of several adverse maternal, fetal, and neonatal health outcomes, including drug poisonings (i.e., drug overdose), [1,2] pregnancy loss, stillbirth, preterm birth, and maternal and neonatal mortality.[3,4] Over the past two decades, the incidence of PSU has increased substantially across North America.[3,5–8] Similarly, rates of unregulated drug poisonings among pregnant and post-partum people have been rising since 2007.[9–12] From 2018–2021 alone, the drug poisoning mortality ratio among pregnant and post-partum people in the United States aged 35–44 more than tripled, from 4.9 to 15.9 per 100,000 pregnant people.[12]

Rates of PSU are particularly high in British Columbia (BC), Canada. Between 2000 and 2020, the incidence of PSU in BC rose from 126 to 247 per 100,000 births.[13] Additionally, a 2021 study concluded that rates of perinatal opioid use disorder (OUD) in BC had risen by 110% since 2001.[6] Results from a 2024 study suggest that overdose is now a major cause of death among pregnant and postpartum people in BC, and these deaths may be underreported.[14]

Comprehensive, integrated care models that incorporate both obstetric and addiction medicine services are essential to improving outcomes for pregnant people who use substances and their infants. These models include coordinated access to opioid agonist therapy (OAT), support for rooming-in practices that promote maternal-infant bonding, and continuity of care during the postpartum period. Such approaches improve maternal and neonatal health outcomes, reduce stigma, and enhance engagement in prenatal and addiction care.[15–19] However, these services are inaccessible for many people in BC, due to the lack of providers experienced in addiction medicine, insufficient OAT provider availability, and the dearth of specialized care for pregnant people who use substances.[5,20] These challenges are especially pronounced in rural and remote locations, and for pregnant people experiencing multiple forms of oppression (e.g., gender and racial minority groups), for whom stigma and the risk of child apprehension are particularly high.[5,20,21] This forced parent-child separation is associated with substantial harms for both parents and children, including increased risk of depression, anxiety, drug use, overdose, and suicidal behavior.[22–25]

All these findings highlight the urgent need for high-quality population-level research that can inform interventions designed to support improved health outcomes among people with PSU and their children. However, PSU has historically been under-researched and underreported due to ethical concerns around studying this marginalized population [26,27] and healthcare system failures to reach and engage individuals with PSU.[28,29] The few studies that do exist are often limited by small sample sizes and a lack of adjustment for confounding.[30]

To address this research gap, a population-based provincial cohort that links mothers with an indication of substance use within 12 months prior to first pregnancy-related healthcare contact and their children was established in 2019. For the purposes of this cohort, the umbrella term "mother" encompasses birthing parents of all gender identities, recognizing that this term unintentionally promotes cisnormativity.[31]

This cohort will serve as a foundation for three research studies. The objectives of these studies are:

1. To evaluate maternal and neonatal health outcomes among mother-infant dyads who accessed specialized inpatient PSU programs during pregnancy.

2. To evaluate the comparative effectiveness of OAT medications and dosing regimens during pregnancy on maternal and neonatal health outcomes.

3. To evaluate the longitudinal relationship between the health of mothers with PSU and the health outcomes of their children over 20 years of follow-up.

## Methods

In BC, all contacts with the healthcare system are collected within population-level databases. The cohort was extracted from 10 linked population-level administrative databases: client roster (demographics), the Medical Services Plan (physician billing), the Discharge Abstract Database (hospitalizations), PharmaNet (drug dispensations), BC Corrections (incarceration in provincial prisons), the National Ambulatory Care Reporting System (emergency department visits), the Perinatal Services BC database (maternal and newborn health for all provincial births), the Social Development and Poverty Reduction (SDPR) database (receipt of housing and income assistance), Vital Statistics (deaths and underlying causes), and the BC Coroners Service database (unregulated drug poisoning deaths) (S1 Table **and** S2 Table). Using unique, de-identified personal health numbers, each mother's health record is also linked to the records of their children.

Using these datasets, we identified all pregnant people with ≥1 indication of substance use (i.e., opioids, alcohol, cannabis, stimulants, sedatives, hypnotics, hallucinogens, or other unspecified substances) between 12 months prior to first pregnancy-related healthcare contact and date of delivery, and who delivered an infant in BC between April 2000–March 2022. As contact with the healthcare system may be relatively infrequent for many people, and in the absence of urine drug testing to confirm substance use, the timeframe of 12 months prior to first pregnancy-related healthcare contact was chosen in an effort to capture ongoing, active substance use at the time of pregnancy. Nicotine and caffeine use was not evaluated in this cohort. Indications of substance use were identified with a case-finding algorithm our team previously developed using Drug Identification Numbers and ICD-9/10-CA codes (S2 Table).[32] Eligible deliveries were those that occurred at ≥20 weeks' gestation (including stillbirths), as mother-infant linkage and birth records are not included in the Perinatal Services BC database prior to this point. As of July 2025, the provincial cohort included 38,670 mothers with an indication of substance use, and 45,823 children born to these individuals (Fig 1).

## Follow-up time

For each individual in the population-based cohort, longitudinal data are available from one year prior to first pregnancy-related healthcare contact until end of data capture (currently December 31, 2022). Using each mother's unique personal health number, individually linked data on demographics and geographic location, physician billing, hospitalization, drug

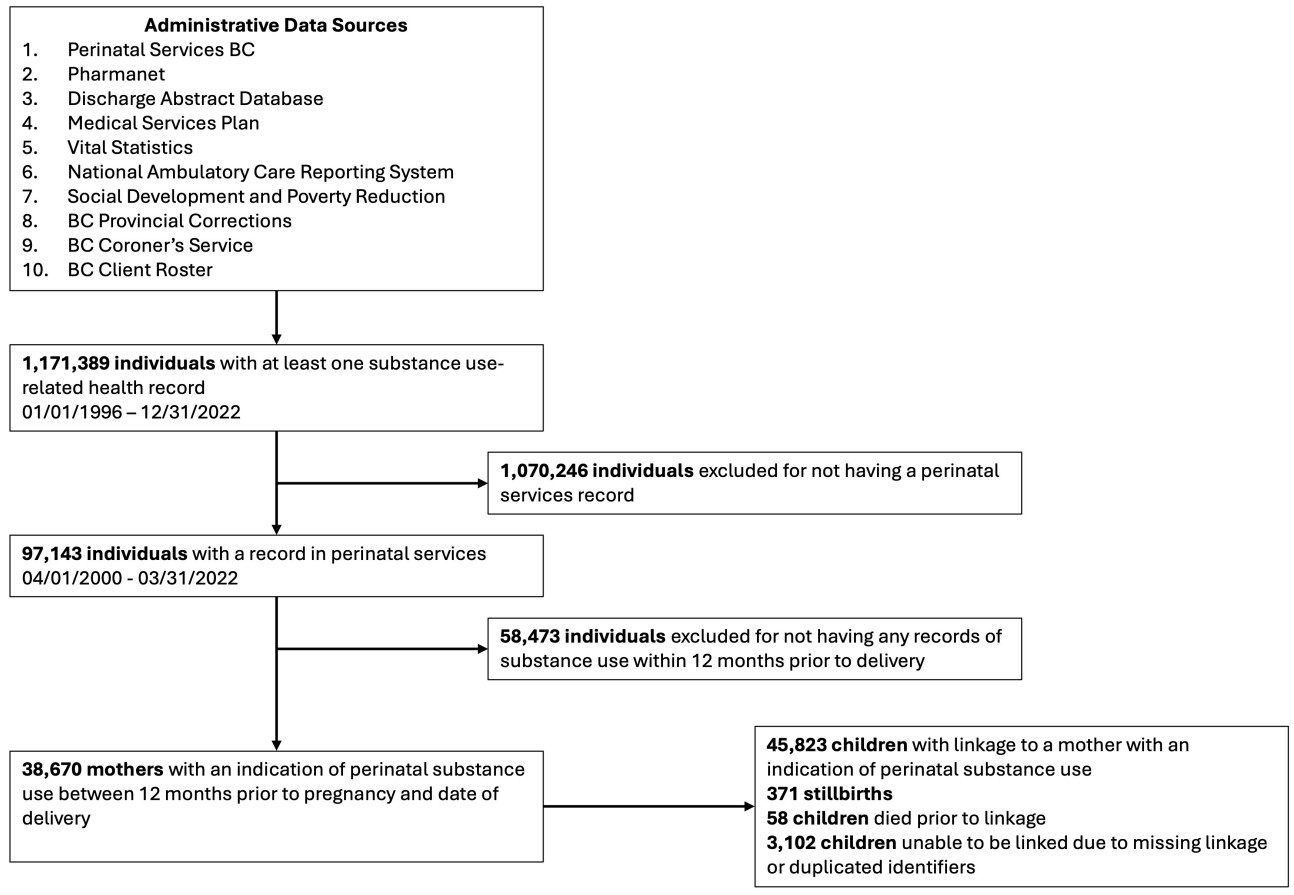

**Fig 1. Cohort Flow Diagram.**

dispensation, and death records are available for the years 1996 – 2022. Data on prenatal care and birth outcomes are available from 2000 – 2022, incarceration, unstable housing, and income assistance data are available from 2010 – 2022, and data on emergency department admissions are available from 2012–2022 (S1 Fig). Data are updated annually.

Individuals were deemed administratively lost to follow-up if they had no record of death at the end of data capture, and no records in the linked health administrative databases for a ≥ 66-month period prior to the end of study follow-up. This cut-off period was empirically determined based on gaps between encounters with the healthcare system captured in our dataset. The chosen cut-off period of 66 months corresponds with the 97.5th percentile in gap times observed in the cohort.[30,32]

Median years of follow-up for mothers in the population-based cohort was 9 (interquartile range [IQR]: 5 – 15) (Table 1). A total of 1,432 mothers (3.7%) and 1,466 children (3.2%) were lost to follow-up and were administratively censored.

## Available data

**Demographic characteristics, clinical characteristics, and healthcare utilization.** Demographic data are captured on age, location, receipt of income assistance, housing stability at the time of delivery, incarceration in provincial institutions, and death. Clinical data are collected for all hospitalizations, physician billing, emergency department visits, maternal and newborn health outcomes, and drug dispensations. These data allow us to measure healthcare utilization and clinical outcomes, both during pregnancy and until the end of data capture (i.e., December 31, 2022), for both

**Table 1. Descriptive statistics of mothers in BC with indication of substance use between one year prior to first pregnancy-related healthcare contact and date of delivery, by substance type (2000/04/01-2022/12/31\*).**

| | Any substances | Opioids | Alcohol only | Cannabis only | Other substances[1] |
|---|---|---|---|---|---|
| | (N = 38,670) | (n = 3,803) | (n = 7,781) | (n = 12,329) | (n = 14,757) |
| **Demographic characteristics** | | | | | |
| Age at delivery, *years* (median, IQR) | 27 (22 –32) | 28 (24 –32) | 28 (23 –33) | 27 (23 –32) | 26 (22 –31) |
| <19 | 2990 (7.7%) | 148 (3.9%) | 554 (7.1%) | 809 (6.6%) | 1,479 (10.0%) |
| 19-24 | 12236 (31.6%) | 1,025 (27.0%) | 2,026 (26.0%) | 3,998 (32.4%) | 5,187 (35.2%) |
| 25-34 | 18560 (48%) | 2,074 (54.5%) | 3,907 (50.2%) | 6,134 (49.8%) | 6,445 (43.7%) |
| >=35 | 4876 (12.6%) | 556 (14.6%) | 1,290 (16.6%) | 1,388 (11.3%) | 1,642 (11.1%) |
| Rural region, n (%) | 9227 (23.9%) | 456 (12.0%) | 2,358 (30.3%) | 3,028 (24.6%) | 3,385 (22.9%) |
| Ever received income assistance, n (%) | 18655 (48.2%) | 3,035 (79.8%) | 2,198 (28.3%) | 5,450 (44.2%) | 7,972 (54.0%) |
| Ever experience unstable housing, n (%) | 1309 (3.4%) | 581 (15.3%) | 71 (0.9%) | 202 (1.6%) | 455 (3.1%) |
| Ever incarcerated, n (%) | 1408 (3.6%) | 728 (19.1%) | 62 (0.8%) | 78 (0.6%) | 540 (3.7%) |
| Died during follow-up, n (%) | 1387 (3.6%) | 358 (9.4%) | 245 (3.2%) | 85 (0.7%) | 699 (4.7%) |
| Years of follow-up included (median, IQR) | 9 (5 –15) | 8 (4 –14) | 11 (6 –16) | 6 (3 –10) | 13 (7 –18) |
| Administrative loss to follow-up, n (%) | 1432 (3.7%) | 107 (2.8%) | 297 (3.8%) | 253 (2.1%) | 775 (5.3%) |
| **Substance use identified from one year prior to first pregnancy-related healthcare contact to date of delivery, n (%)** | | | | | |
| Alcohol use | 12857 (33.2%) | 743 (19.5%) | 7,781 (100%) | ----- | 4,333 (29.4%) |
| Cannabis use | 16994 (43.9%) | 770 (20.3%) | ----- | 12,329 (100%) | 3,895 (26.4%) |
| Stimulant use | 5534 (14.3%) | 1,435 (37.7%) | ----- | ----- | 4,099 (27.8%) |
| Other substances | 14643 (37.9%) | 2,936 (77.2%) | ----- | ----- | 11,707 (79.3%) |
| **OAT during pregnancy, n (%)** | ----- | 2,106 (55.4%) | ----- | ----- | ----- |
| 1st trimester | ----- | 1,547 (40.7%) | ----- | ----- | ----- |
| 3rd trimester | ----- | 1,892 (49.8%) | ----- | ----- | ----- |
| **Comorbidities at time of delivery, n (%)** | | | | | |
| Chronic pain, non-cancer | 16094 (41.6%) | 1,983 (52.1%) | 2,886 (37.1%) | 5,297 (43.0%) | 5,928 (40.2%) |
| Gestational diabetes | 2296 (5.9%) | 173 (4.6%) | 428 (5.5%) | 995 (8.1%) | 700 (4.7%) |
| Carlson comorbidity index > 1 within past 4 years[2] | 1269 (3.3%) | 349 (9.2%) | 161 (2.1%) | 205 (1.7%) | 554 (3.8%) |
| **Mental health disorders at time of delivery[3], n (%)** | 23523 (60.8%) | 3,060 (80.5%) | 3,745 (48.1%) | 7,392 (60.0%) | 9,326 (63.2%) |
| Anxiety disorders | 13868 (35.9%) | 2,296 (60.4%) | 2,088 (26.8%) | 4,006 (32.5%) | 5,478 (37.1%) |
| Non-major depression or other mood disorders | 11826 (30.6%) | 1,869 (49.2%) | 1,695 (21.8%) | 3,284 (26.6%) | 4,978 (33.7%) |
| Psychotic disorder[4] | 907 (2.3%) | 237 (6.2%) | 80 (1.0%) | 137 (1.1%) | 453 (3.1%) |
| Behavioural disorder | 2532 (6.5%) | 389 (10.2%) | 265 (3.4%) | 881 (7.2%) | 996 (6.8%) |
| Personality disorder | 1997 (5.2%) | 458 (12.0%) | 194 (2.5%) | 365 (3.0%) | 980 (6.6%) |
| Other disorder[5] | 2151 (5.6%) | 369 (9.7%) | 296 (3.8%) | 589 (4.8%) | 897 (6.1%) |
| Serious mental illness | 3365 (8.7%) | 600 (15.8%) | 447 (5.7%) | 843 (6.8%) | 1,475 (10.0%) |
| Major depressive disorder | 3102 (8%) | 546 (14.4%) | 425 (5.5%) | 760 (6.2%) | 1,371 (9.3%) |
| Bipolar disorder | 953 (2.5%) | 184 (4.8%) | 91 (1.2%) | 193 (1.6%) | 485 (3.3%) |
| Schizophrenia | 545 (1.4%) | 139 (3.7%) | 42 (0.5%) | 81 (0.7%) | 283 (1.9%) |
| Indication of self-harm | 3234 (8.4%) | 602 (15.8%) | 475 (6.1%) | 587 (4.8%) | 1,570 (10.6%) |
| **Healthcare utilization within 12 months prior to delivery, n (%)** | | | | | |
| Any hospitalizations | 8433 (21.8%) | 1,366 (35.9%) | 1,397 (18.0%) | 1,808 (14.7%) | 3,862 (26.2%) |
| Any emergency department visits | 20304 (52.5%) | 2,323 (61.1%) | 3,664 (47.1%) | 7,256 (58.9%) | 7,061 (47.9%) |

*(Continued)*

**Table 1.** (Continued)

| | Any substances | Opioids | Alcohol only | Cannabis only | Other substances[1] |
|---|---|---|---|---|---|
| **Pregnancy outcomes at delivery** | | | | | |
| Stillbirth | 257 (0.7%) | 38 (1.0%) | 44 (0.6%) | 62 (0.5%) | 113 (0.8%) |
| Caesarean section | 11455 (29.6%) | 1,290 (33.9%) | 2,166 (27.8%) | 3,820 (31.0%) | 4,179 (28.2%) |

1. Any indication of perinatal substance use excluding opioids but including alcohol or cannabis when other substances were also identified. 2. Past four years is provided since historical follow-up begins January 1st 1996, but perinatal data starts from April 1st 2000. 3. Specific conditions will add up to more than total as 10,481 (27.1%) of mothers had more than 1 mental health disorder. 4. Excluding schizophrenia. 5. Includes developmental disorders and other childhood onset disorders, personality and psychotic disorders

Abbreviations: OAT: Opioid agonist treatment.

All variables are defined in Supplementary S1 Table and S2 Table.

*Birth data available until March 31, 2022.

mothers and their children. Available data include health records related to acute and chronic conditions, mental health diagnoses, and substance use disorder diagnoses.

**Novel measurements.** The unique linkages between mothers and their children over time allows us to measure the ways in which maternal health outcomes and life events may affect child health outcomes. For example, in our cohort, mother and child separations are only captured if the separation occurs at discharge from hospital. Later separations can be inferred based on the geographic location of the mother and child in their individual health records (available in multiple component datasets; subject to evaluations of concordance) or by assessing whether a mother-child pair are attached to the same social assistance record. This information can then be used to evaluate the association between such separations and adverse physical and mental health outcomes for both mother and child. Importantly, our data do not distinguish between separations where a child is placed in kinship care (i.e., with a family member) versus foster care, nor can we distinguish between separations made by choice versus state apprehension. Similarly, although direct linkages to non-birthing parents (e.g., fathers, stepparents) are unavailable, in instances where the non-birthing parent is part of the broader provincial substance use cohort, [32] these individuals are linked to their children or the mothers via social assistance records. The longitudinal nature of our data also allows us to evaluate trends in maternal and child health outcomes over time, enabling the exploration of the impact of specific events, such as the COVID-19 pandemic, the end of birth alerts in 2019, and the introduction of fentanyl into the unregulated drug supply.

## Ethical approval

This study has been determined to meet the criteria for exemption per Article 2.5 of the 2018 Tri-Council Policy Statement: Ethical Conduct for Research Involving Humans. Study databases have been made available by the BC Ministries of Health and Mental Health and Addiction as part of the provincial opioid overdose public health emergency response. All data in the cohort are de-identified, and patient consent for the secondary use of these data was not required.

## Results

### Demographic and clinical characteristics

The demographic and clinical characteristics of mothers and children in the population-based cohort are described in **Table 1** and **Table 2**, respectively. Of the 38,670 mothers with an indication of substance use between 12 months prior

**Table 2. Descriptive statistics of children born to mothers in BC with indication of substance use between 12 months prior to first pregnancy-related healthcare contact and time of delivery by substance type (2000/04/01-2022/12/31*).**

| | Any substances | Opioids | Alcohol only | Cannabis only | Other substances[1] |
|---|---|---|---|---|---|
| | (N = 45,823) | (N = 5,042) | (N = 8,661) | (N = 15,200) | (N = 16,920) |
| Age at end of data capture, *years*** | 9 (4 –14) | 8 (4 –13) | 10 (6 –15) | 6 (3 –9) | 13 (7 –17) |
| Administrative loss to follow-up, n (%) | 1,466 (3.2%) | 113 (2.2%) | 311 (3.6%) | 295 (1.9%) | 747 (4.4%) |
| Death during follow-up, n (%) | 468 (1.0%) | 84 (1.7%) | 90 (1.0%) | 78 (0.5%) | 216 (1.3%) |
| **Mother's death during follow-up, n (%)** | 1,524 (3.3%) | 377 (7.5%) | 277 (3.2%) | 115 (0.8%) | 755 (4.5%) |
| Death within first year of life | 133 (0.3%) | 52 (1.0%) | 13 (0.2%) | 11 (0.1%) | 57 (0.3%) |
| Death prior to age 5 | 531 (1.2%) | 174 (3.5%) | 69 (0.8%) | 56 (0.4%) | 232 (1.4%) |
| **Ever identified as separated from mother, n (%)** | 27,514 (60.0%) | 3,967 (78.7%) | 4,553 (52.6%) | 6,979 (45.9%) | 12,015 (71.0%) |
| Separated from hospital discharge | 1,927 (4.2%) | 609 (12.1%) | 111 (1.3%) | 107 (0.9%) | 1,110 (6.5%) |
| Separation identified from SDPR[2] | 13,552 (29.6%) | 2,913 (57.8%) | 1,723 (19.9%) | 2,338 (15.4%) | 6,578 (38.9%) |
| Separation identified from mismatched location[2] | 24,812 (54.1%) | 3,427 (68.0%) | 4,211 (48.6%) | 6,196 (40.8%) | 10,978 (64.9%) |
| Separation due to incarceration of mother | 2,736 (6.0%) | 801 (15.9%) | 240 (2.8%) | 176 (1.2%) | 1,519 (9.0%) |
| **Any mental health disorder diagnosis[3], n (%)** | 12,053 (26.3%) | 1,749 (34.7%) | 2,039 (23.5%) | 2,230 (14.7%) | 6,035 (35.7%) |
| Anxiety, depression or other mood disorder[4] | 5,848 (12.8%) | 635 (12.6%) | 1,168 (13.5%) | 554 (3.6%) | 3,491 (20.6%) |
| Developmental disorder | 3,128 (6.8%) | 614 (12.2%) | 461 (5.3%) | 639 (4.2%) | 1,414 (8.4%) |
| Behavioral disorder | 6,408 (14%) | 1,011 (20.1%) | 929 (10.7%) | 1,196 (7.9%) | 3,272 (19.3%) |
| Other mental health condition[5] | 3,310 (7.2%) | 542 (10.8%) | 511 (5.9%) | 688 (4.5%) | 1,569 (9.3%) |
| Serious mental health disorder[6] | 1,006 (2.2%) | 108 (2.1%) | 192 (2.2%) | 40 (0.3%) | 666 (3.9%) |
| Incidence of self-harm | 1,189 (2.6%) | 146 (2.9%) | 209 (2.4%) | 75 (0.5%) | 759 (4.5%) |
| Incidence of child maltreatment or peer violence | 7,047 (15.4%) | 910 (18.1%) | 1,393 (16.1%) | 1,681 (11.1%) | 3,063 (18.1%) |
| **Psychiatric prescription medications, n (%)** | | | | | |
| Benzodiazepines | 1,911 (4.2%) | 205 (4.1%) | 419 (4.8%) | 215 (1.4%) | 1,072 (6.3%) |
| Antipsychotics or antimanic | 1,711 (3.7%) | 227 (4.5%) | 280 (3.2%) | 134 (0.9%) | 1,070 (6.3%) |
| SSRIs or other antidepressants | 3,907 (8.5%) | 404 (8.0%) | 791 (9.1%) | 223 (1.5%) | 2,489 (14.7%) |
| Stimulants | 4,972 (10.9%) | 782 (15.5%) | 764 (8.8%) | 825 (5.4%) | 2,601 (15.4%) |
| **Acute care utilization, n (%)** | | | | | |
| Any hospitalizations | 18,184 (39.7%) | 2,306 (45.7%) | 3,606 (41.6%) | 4,389 (28.9%) | 7,883 (46.6%) |
| Any emergency department visits** | 5 (2 –11) | 6 (2 –11) | 6 (2 –12) | 4 (1 –8) | 7 (2 –14) |
| **Birth outcomes** | | | | | |
| Gestational age** | 39 (37 –40) | 38 (36 –39) | 39 (38 –40) | 39 (37 –40) | 39 (37 –40) |
| Preterm, n (%) | 7,010 (15.3%) | 1,317 (26.1%) | 1,028 (11.9%) | 1,985 (13.1%) | 2,680 (15.8%) |
| Birth weight (grams)** | 3312 (2943–3670) | 3096 (2700–3470) | 3424 (3075–3760) | 3320 (2965–3660) | 3310 (2930–3670) |
| Low birth weight, n (%) | 4,154 (9.1%) | 844 (16.7%) | 528 (6.1%) | 1,199 (7.9%) | 1,583 (9.4%) |
| Small for gestational age and related conditions, n (%) | 8,232 (18.0%) | 1,734 (34.4%) | 1,112 (12.8%) | 2,392 (15.7%) | 2,994 (17.7%) |

*Birth data available until March 31, 2022. **Median (1st quartile – 3rd quartile).

1. Any indication of mother's substance use excluding opioids but including alcohol or cannabis when other substances were also identified

2. Multiple methods were used to estimate the number of separations in the cohort; these estimates should not be added together. Only one instance of separation was counted per child.

3. Specific conditions will add up to more than total as 5,020 (11.0%) of children had more than 1 mental health disorder.

4. Excluding major depressive disorder

5. Includes other childhood onset disorders, personality and psychotic disorders

6. Includes major depressive disorder, bipolar disorder, and schizophrenia

to first pregnancy-related healthcare contact and time of delivery, 31.9% (n = 12,329) used cannabis only. A total of 3,803 mothers (9.8%) had an indication of opioid use during this time period.

The median age at delivery among mothers was 27 years (interquartile range [IQR]: 22–32). Nearly half of mothers (n = 18,655, 48.2%) had received income assistance prior to the birth of their first child in the cohort. Across categories of prenatal substance exposure, individuals with an indication of prenatal opioid use had the highest rates of unstable housing (n = 581, 15.3%) and receipt of income assistance (n = 3,035, 79.8%) at any point prior to delivery, and the highest rates of diagnoses for chronic pain, comorbidities, emergency room visits, and hospitalizations during the follow-up period (Table 1).

Among the 45,823 children in the population-based cohort, 15.3% (n = 7,010) were born preterm (i.e., < 37 weeks' gestation), and 9.1% (n = 4,154) were born with low birthweight (i.e., < 2,500g). These rates are higher than those identified among the broader BC population (preterm birth: 11.6%; low birthweight: 4.9%).[33] These adverse outcomes were particularly common among children born to mothers with an indication of opioid use (Table 2). A total of 4.2% (n = 1,927) were separated from their mothers at hospital discharge. Using the SDPR database, 29.6% (n = 13,552) of children experienced separation from their mothers during ≥1 calendar month over the follow-up period (Table 2).

## Concurrent mental health conditions

Prior to delivery, major depressive disorder was identified in 8.0% of mothers overall (n = 3,365). Indications of self-harm were documented among 8.4% of mothers (n = 3,234). Mental health disorders were present among 26.3% of children over the course of follow-up (n = 12,053) and over 40% of children with ≥16 years of follow-up (Fig 2). The most common diagnoses in children were anxiety, depression, or other mood disorders (n = 5,848, 12.8%) (Fig 2).

## Maternal mortality

During the follow-up period, 1,387 maternal deaths were recorded, representing 3.6% of the total cohort. From 2004–2022, the rate of maternal deaths per 1000 people increased from 2.0 to 5.6, with a substantial increase observed starting in 2016, when a public health emergency was declared in BC due to escalating numbers of drug poisoning-related deaths caused by the saturation of fentanyl in the unregulated drug supply (Fig 3). Indeed, drug-related causes were a major contributor to maternal mortality in this cohort, with a noticeable increase in the proportion of deaths attributed to drug-related causes beginning in 2016 (Fig 3). Notably, a total of 358 deaths occurred among mothers with an indication of opioid use during the follow-up period, representing 9.4% of these mothers. A total of 133 children (0.3%) experienced the death of their mother within the first year of life, and 531 (1.2%) experienced this loss within the first five years of life.

## Discussion

In this cohort profile, we detail construction of a population-based cohort of mothers with SUD and their children, describing key characteristics and the breadth of data available.

The major strength of this cohort is its longitudinal, population-based design, which allows us to capture data spanning over two decades from all individuals in the province with a record of prenatal substance use. In addition, as the cohort is updated annually, our analyses can be updated periodically to explore long-term maternal-child health outcomes, and to investigate the impact of factors such as policy changes, changes in the unregulated drug supply, or acute public health events (e.g., COVID-19).

Our ability to link data from ten health administrative databases also offers detailed insights into the healthcare system's successes or failures to provide cohort members with healthcare and social services, and how this engagement impacts clinical outcomes over time. The ability to link maternal-child records permits investigation of how systemic health inequities shape maternal experiences of disadvantage (e.g., mental health disorders, inadequate housing, forced separation) and how these experiences may impact child health and development.

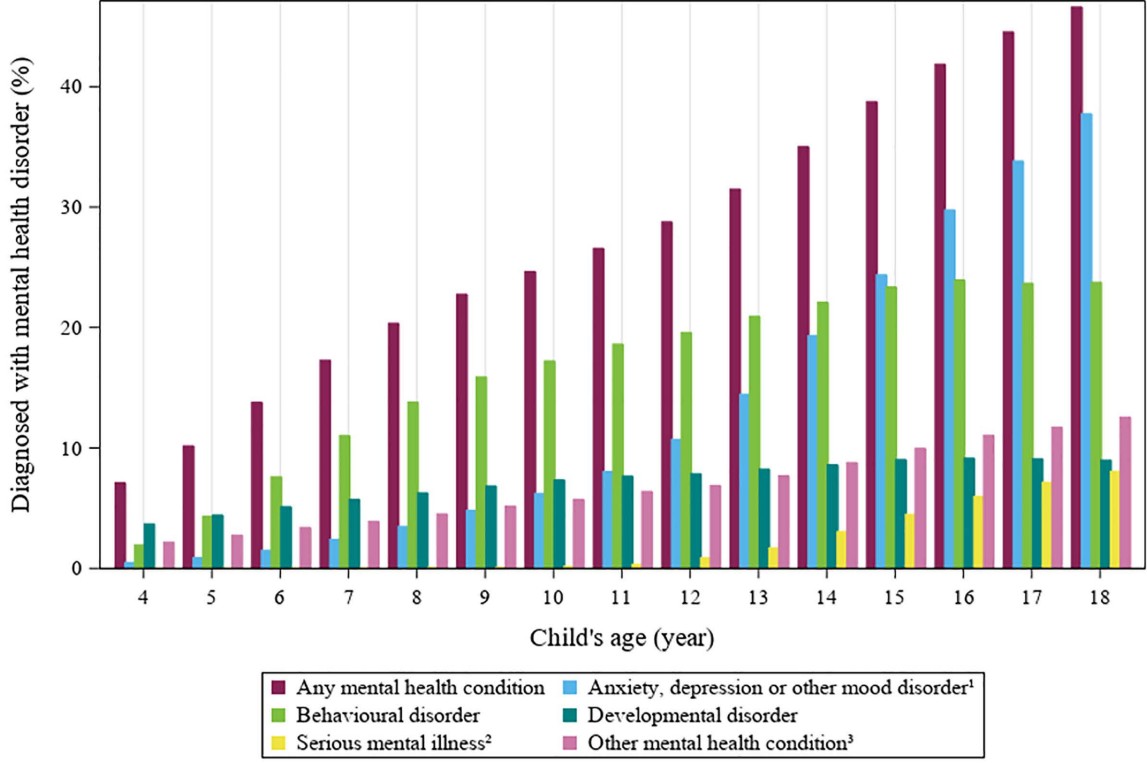

**Fig 2. Mental health diagnoses among children born to mothers with perinatal substance use from ages 4 to 18 1.** Includes depression (not major depressive disorder), mood and anxiety disorder. 2. Includes major depressive disorder, bipolar disorder, and schizophrenia. 3. Includes other childhood onset disorders, personality and psychotic disorders.

## Limitations

There are also several limitations. As health administrative datasets are primarily intended for reimbursement rather than research purposes, some important information cannot be captured, including health events for which healthcare was not received, as well as social factors (e.g., socioeconomic status, employment, adverse childhood events). Future research should aim to address these gaps through prospective, participatory, qualitative research.

Importantly, the datasets do not contain direct observations of substance use, either through self-report or urine drug testing. This means that individuals without an indication of substance use in their health administrative records may be missing from the cohort. Moreover, our classifications of substance use disorders and administrative loss to follow-up may result in misclassification. We will consider alternative thresholds to assess the robustness of results to these thresholds. Our data are also not linked to data from the BC Ministry of Child and Family Development, meaning we are unable to identify children separated by state apprehensions. There were also 3,102 children who could not be linked to their mothers. These children included twins whose records could not be distinguished from one another, and children with missing linkage information. This limitation will be addressed in a forthcoming data update.

With the exception of individuals who are in the broader substance use cohort, our population-based cohort does not currently contain information on fathers or other non-birthing parents. This limitation will also be mitigated in a future data update.

There are also limitations within the datasets. The PharmaNet database does not capture medications dispensed in hospital, and is missing costing information for claims paid by the Canadian federal government, as well as antiretroviral

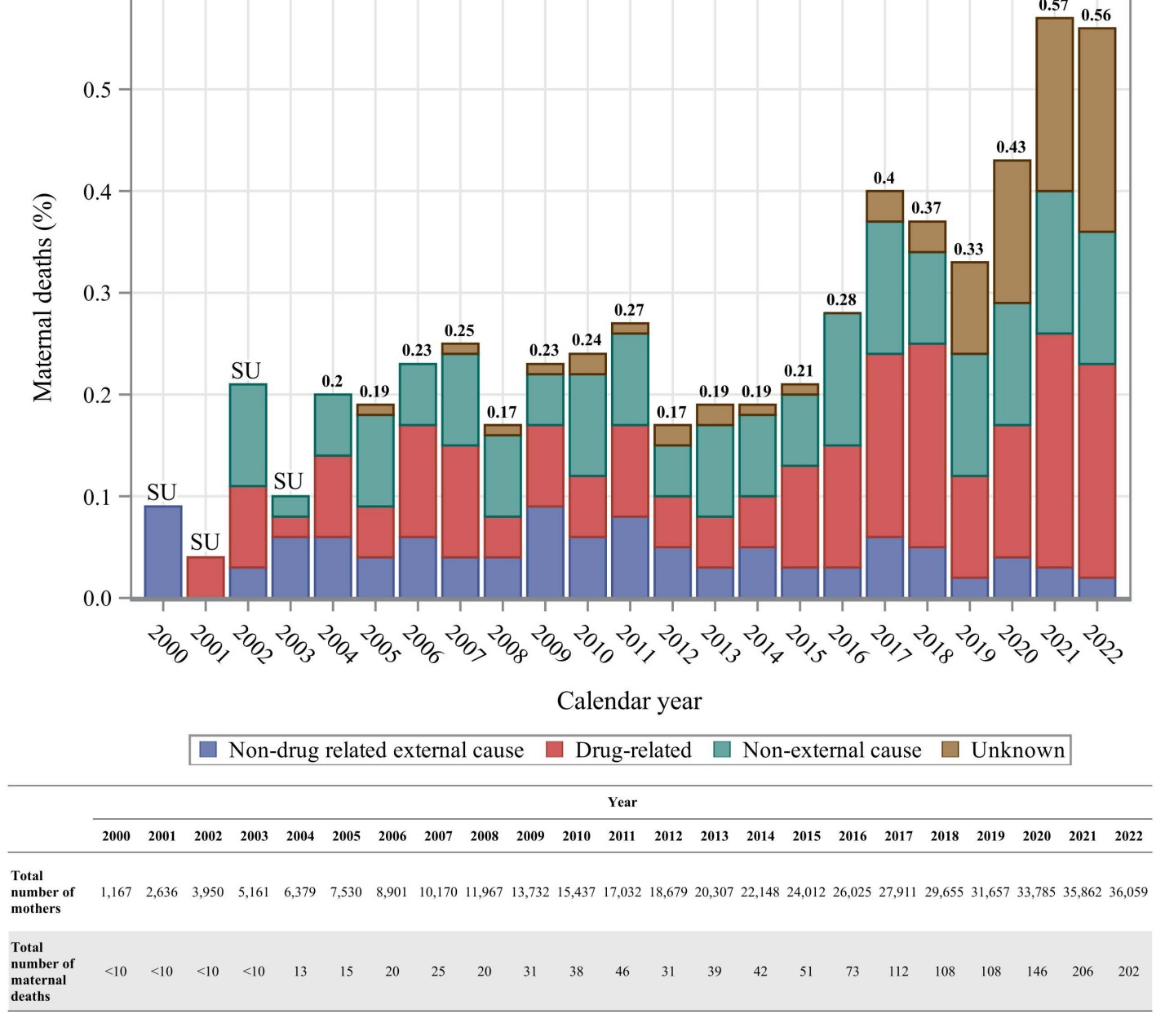

**Fig 3. Maternal deaths between 2000/04/01–2022/12/31.** The increase in the proportion of deaths due to unknown causes since 2019 is due to a lag in data availability.

medications.[34] Similarly, the Medical Services Plan database do not capture billing records under the Alternative Payment Plan, most often associated with community health centres.[35]

Lastly, many harm reduction services, which are an important part of the continuum of care for individuals who use substances, [36] do not require individuals to provide identifying information in order to use these services. As a result, these interactions are not captured in provincial databases. Further linkage to community and local services is necessary to characterize health care service utilization, including the use of peer support services.

## Conclusion

The growing population of mothers with substance use disorder represent a critical need for the province of British Columbia. This population-based retrospective cohort will be used to inform service design and otherwise advocate for client needs.

## Supporting information

**S1 Fig. Data collection timeline for each provincial administrative database (01/01/1996-31/12/2022).** MSP, Medical Services Plan; DAD, Discharge Abstract Database; BCVS, BC Vital Statistics; PNET, Pharmanet; PSBC, Perinatal Services BC; SDPR, BC Social Development and Poverty Reduction; NACRS, National Ambulatory Care Reporting System; BCCS, BC Coroners Service; BC Corrections, BC Provincial Corrections.
(DOCX)

**S1 Table. Description of ten provincial administrative database in British Columbia, 2000–2022.** Abbreviations: DIN: Drug Identification Number; PIN: Product Identification Number; † Coding structures used by the Canadian Institute of Health Information.
(DOCX)

**S2 Table. Case finding algorithm of substance use, comorbidities, other conditions, and social determinants of health.** AHFS: American Hospital Formulary Service by the American Society of Health-System Pharmacists; BCPDR: British Columbia Perinatal Data Registry; DAD: Discharge Abstract Database (hospitalizations); DIN: drug identification number in PharmaNet (drug dispensations); ICD-9-CA: International Classification of Diseases, Ninth Revision, Canada. ICD-10-CA: International Statistical Classification of Diseases and Related Health Problems, Tenth Revisions, Canada; MSP: Medical Services Plan; NACRS: National Ambulatory Care Reporting System (emergency department visits); PIN: product identification number in PharmaNet (drug dispensations); SDPR: Social Development and Poverty Reduction; VS: Vital Statistics database in British Columbia (death records). a Diacetylmorphine or hydromorphone with some restrictions based on prescriber, dispensing pharmacy and/or date. *Pharmacare Plan C (Income Assistance) provides full coverage of eligible prescription costs for B.C. residents receiving benefits and income assistance through the Ministry of Social Development and Poverty Reduction, or in the care of, or in an agreement with Ministry of Children and Family Services for children and youth.
(DOCX)

## Acknowledgments

We gratefully acknowledge the British Columbia Ministry of Health for the acquisition of provincial datasets and the organizations included in this work serving the community. This work was conducted on the unceded, occupied, traditional and ancestral lands of the Coast Salish Peoples, including the snʊʼnɛɪməxʷ (Snuneymuxw), xʷməθkʷəy̓əm (Musqueam), Skwxwú7mesh (Squamish), and Səl̓ílwətaʔ (Tsleil-Watuth) Nations. Data were derived from the lands of the 204 distinct First Nations in what is colonially known as British Columbia. We acknowledge the disproportionate impact of the toxic drug crisis and the harms caused by colonial systems including healthcare and child welfare on Indigenous Peoples across Turtle Island.

## Author contributions

**Conceptualization:** Bohdan Nosyk.

**Formal analysis:** Noah Katsuno, Megan Kurz.

**Funding acquisition:** Bohdan Nosyk.

**Methodology:** Lindsay A Wilson.

**Writing – original draft:** Lindsay A Wilson.

**Writing – review & editing:** Lindsay A Wilson, Noah Katsuno, Megan Kurz, Nicole Catherine, Shannon Joyce, Brittany Barker, Brittany B Dennis, Sarah Moreheart, Lianping Ti, Jane A Buxton, Bohdan Nosyk.

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
