## [Decision Letter · Decision Letter 0]

18 Feb 2026

PONE-D-25-53902Cohort profile: mothers who use substances and their children in British Columbia, CanadaPLOS One

Dear Dr. Nosyk,

Thank you for submitting your manuscript to PLOS ONE. After careful consideration, we feel that it has merit but does not fully meet PLOS ONE’s publication criteria as it currently stands. Therefore, we invite you to submit a revised version of the manuscript that addresses the points raised during the review process.

Thank you for your patience with the peer review. The manuscript is timely and well-written. I have reached a final decision of "minor revision".

We look forward to receiving your revised manuscript.

Kind regards,

David T. Zhu

Academic Editor

PLOS One

Journal Requirements:

Reviewers' comments:

Reviewer's Responses to Questions

Comments to the Author

1. Is the manuscript technically sound, and do the data support the conclusions?

Reviewer #1: Partly

Reviewer #2: Yes

2. Has the statistical analysis been performed appropriately and rigorously? 

Reviewer #1: No

Reviewer #2: N/A

3. Have the authors made all data underlying the findings in their manuscript fully available?

Reviewer #1: Yes

Reviewer #2: Yes

4. Is the manuscript presented in an intelligible fashion and written in standard English?

Reviewer #1: Yes

Reviewer #2: Yes

5. Review Comments to the Author

Reviewer #1: Reviewer Comments for Manuscript PONE-D-24-45499R3

This manuscript contains grammatical errors, awkward phrasing, and inconsistent tense usage that reduce clarity and readability. Several sentences are overly long or redundant, which weakens the logical flow of the argument. I recommend a thorough language edit to:

• Correct grammar and typographical errors.

• Ensure consistent use of past tense in the Methods and Results sections, and present tense in the Introduction and Discussion…(e.g., “effective coverage,” “crude coverage”)

• Shorten or restructure overly long sentences to improve conciseness and readability. Minor typographical errors and redundancies should be carefully polished before publication.

Suggestion: The manuscript would benefit significantly from thorough proofreading by a native English speaker or professional editing service to improve clarity and adherence to academic writing standards.

2. Abstract

• The abstract is informative but needs refinement for clarity.

• The study period is confusing: the abstract mentions June–July 2023 (1 month), while the methods section refers to September 2021–August 2022 (1 year). This inconsistency must be corrected. You have to remove … “using secondary data of mothers who received antenatal care from September 01, 2021, to August 30, 2022”. You have also change design into “retrospective cross-sectional studies”. Change…”bi variable logistic regression” → “bivariable logistic regression”;

• The results section of the abstract should align with the operational definition of effective coverage. For example, if screening for gestational diabetes is required, how does the reported 25.9% coverage reconcile with only 8.2% receiving that service?

3. Introduction

• The introduction is well contextualized, but some sentences are repetitive and could be streamlined.

• The rationale for focusing on effective coverage rather than crude coverage is well stated, but the flow could be improved by reducing redundancy.

• Consider clarifying the conceptual framework of effective coverage (need, use, quality) more succinctly.

• Suggestion: Consider adding a sentence on why West Gojjam Zone specifically was chosen—e.g., known gaps in ANC quality, high maternal mortality, or representative setting.

• Question: The authors state that “none have examined EC of ANC in our region especially in my study setting.” Were there any sub-national or facility-based ANC quality assessments that could be referenced for context?

Methods

Study design and period

The description of the study design is somewhat confusing. The manuscript currently states: “A facility-based analytical cross-sectional study design was conducted in West Gojjam public hospitals from June 30 to July 30, 2023, utilizing secondary data from mothers who received antenatal care between September 1, 2021, and August 30, 2022.”

• Since the data were extracted retrospectively from ANC charts, the design would be more accurately described as a retrospective cross-sectional study rather than “analytical cross-sectional.”

• The phrase “utilizing secondary data from mothers who received antenatal care between September 1, 2021, and August 30, 2022” is redundant here, because the study population and source of data are already explained elsewhere.

• I recommend that the authors simplify this section to report only the study design and data collection period, while clearly noting that secondary data were used. For example: “A facility-based retrospective cross-sectional study was conducted in West Gojjam public hospitals. Data were collected between June 30 and July 30, 2023, from ANC charts covering the period September 1, 2021, to August 30, 2022.”

Sampling Size Calculation

I appreciate that the authors calculated the sample size for both objectives (coverage and associated factors). However, including the full formula in the manuscript is not necessary, since you have already stated that the single population proportion formula was used.

I recommend simplifying this section by:

• Clearly stating that the sample size was calculated for both objectives.

• Indicating that the largest sample size obtained was selected as the final sample size for the study.

• Removing the redundant formula presentation, while keeping the assumptions (confidence level, margin of error, prevalence, design effect, non-response rate) clearly described.

This will make the Methods section more concise and reader friendly, while still transparent about how the sample size was determined.

Study Design and Sampling:

• The multi-stage random sampling is appropriate, but the description of selecting 3 out of 8 hospitals (38%) by “lottery” is vague. Were all hospitals eligible? Was stratification considered?

• Suggestion: Clarify whether the selection was truly random or purposive, and justify the sample size allocation.

Why were only 3 hospitals selected? Was power considered at the hospital level?

Why was a design effect of 1.5 applied? Was clustering expected, and if so, at what level? Please justify this choice.

Measurement of Effective Coverage:

The definition aligns with WHO recommendations, but the operationalization—“received all the WHO recommended interventions at least once”—may overlook frequency and timing of interventions, which is acknowledged as a limitation.

Suggestion: Briefly note how missing data on intervention timing was handled.

How was “adequately trained human power” assessed? Was it based on presence per shift, or per patient load?

Clarify whether data were extracted only from charts or supplemented with interviews.

Line 181-182. Avoid----remove…”The questionnaire was prepared in English and

182 pretested among 5% (34) of study population before the actual data collection”…this repetition/The author already included in data quality control.

Data Analysis:

The use of p<0.25 for inclusion in multivariable analysis is acceptable but should be justified given the sample size.

Suggestion: Consider also using variance inflation factors (VIF) to check multicollinearity, especially among obstetric/gynecological variables.

Question: Why were hospital-level factors (e.g., equipment, staffing) not included in the regression model?

Results:

Tables are comprehensive but could be better formatted. For example, Table 5 is split across pages Comment on Definition and Results of Effective Coverage

The manuscript defines effective coverage of ANC interventions as: pregnant women who attended four or more ANC visits and received all WHO recommended interventions at least once during their ANC follow up period.

However, the results presented raise important concerns:

• For several interventions, the proportion of women receiving them “at least once” is far below 25.9%. For example, only 8.2% received antenatal ultrasound and 8.2% were screened for gestational diabetes mellitus.

• If all WHO recommended interventions are required for effective coverage, then the reported 25.9% effective coverage seems inconsistent with these low uptake rates.

• The table also mixes “at each visit” and “at least once” indicators, which makes interpretation difficult. For example, gestational diabetes screening is reported as 25.9% “at each visit” but only 8.2% “at least once.” This needs clarification.

• The operational definition should be applied consistently: if “all WHO interventions at least once” is the criterion, then the effective coverage proportion should reflect the lowest uptake intervention (e.g., 8.2%).

Recommendations:

1. Clarify the operational definition of effective coverage — is it “all WHO interventions at least once” or “some interventions at least once”?

2. Ensure consistency between the definition and the reported results.

3. Revise the Results section to avoid contradictory figures (e.g., 25.9% vs. 8.2%).

4. Consider presenting both crude coverage (ANC 4+) and effective coverage (ANC 4+ plus all WHO interventions) separately, with clear explanation.

Discussion:

The discussion appropriately contextualizes findings within existing literature and offers plausible explanations for discrepancies.

To strengthen the impact of your manuscript, please revise the Discussion to move beyond restating the Results. Instead, focus on interpreting the clinical and public health significance of your key findings. For each major result, please address:

1. What does this mean for practice? (e.g., How should the very low rate of Gestational Diabetes Mellitus screening change clinical protocols or health worker training?)

2. What are the policy implications? (e.g., Does the finding that women with prior complications receive better coverage suggest a need to reallocate resources or change messaging to reach low-risk women?)

3. How can these findings inform action? Conclude with targeted, practical recommendations for health system managers and policymakers in your setting."

Question: The finding that prior complications increase effective coverage suggests that risk-based care is occurring. Does this imply that low-risk women are being neglected? This has important programmatic implications.

Reviewer #2: Review of "Cohort Profile: Mothers who use substances and their children in British Columbia, Canada"

16 February 2026

Lines 71-84

The authors might consider adding a brief description of substance treatment that exists outside the medical model, e.g. 12 Step and other peer-to-peer recovery programs, for clarity, and whether this is part of the studies being planned. Notably, this type of peer-to-peer delivery of services is a significant support for many people trying to recover from substance use disorders.

Line 133

Consider changing "Tobacco use was not evaluated." to "Tobacco and caffeine use was not evaluated." or something similar, for additional clarity, and to help destigmatize drug use by highlighting that the majority of Canadian adults use substances, e.g. caffeine.

Table 2

The table has two asterisked footnotes. Consider switching one of these to a different symbol, for clarity.

Lines 278-279

Consider including indigenous and immigrant status in the list of social factors that may affect health outcomes, for clarity, and to highlight these differentials.

Lines 301-304

The authors might consider adding a description of difficulty in including peer-to-peer recovery support below the text related to harm reduction services, for clarity.

Supplemental Table 2

In the footnote, consider changing "... , or In the care of ..." to "... , or in the care of ..." to correct a typographical error.

6. PLOS authors have the option to publish the peer review history of their article (what does this mean?). If published, this will include your full peer review and any attached files.

Do you want your identity to be public for this peer review? For information about this choice, including consent withdrawal, please see our Privacy Policy.

Reviewer #1: No

Reviewer #2: No

---

## [Author Response · Author response to Decision Letter 1]

2 Apr 2026

Journal Requirements:

RESPONSE: Done, as requested.

RESPONSE: Unfortunately, de-identified data are not available for public access, according to the British Columbia Ministry of Health and the UBC Research Ethics Board. We have added this information in the disclaimer, page 2, line 29. A link to access data has been added on line 33.

RESPONSE: We have now moved the ethical approval subsection to the end of the methods section, page 15, line 201.

RESPONSE: Done, as requested.

RESPONSE: Done, as requested.

RESPONSE: Done, as requested.

Reviewers' comments:

Reviewer's Responses to Questions:

Comments to the Author

1. Is the manuscript technically sound, and do the data support the conclusions?

Reviewer #1: Partly

Reviewer #2: Yes

2. Has the statistical analysis been performed appropriately and rigorously?

Reviewer #1: No

Reviewer #2: N/A

3. Have the authors made all data underlying the findings in their manuscript fully available?

Reviewer #1: Yes

Reviewer #2: Yes

4. Is the manuscript presented in an intelligible fashion and written in standard English?

Reviewer #1: Yes

Reviewer #2: Yes

5. Review Comments to the Author

Comments from reviewer #1:

Comments for Manuscript PONE-D-25-53902

I would like to commend the authors for their timely and important work in establishing a population-based cohort of mothers who use substances and their children in British Columbia. The study addresses a significant public health issue, and the cohort will serve as a valuable resource for future research aimed at improving maternal and child health outcomes in this vulnerable population. I also appreciate the editor for the opportunity to review this manuscript.

General Comments

The authors should carefully follow the PLOS ONE manuscript preparation guidelines. The current version lacks several required structural elements:

• The Abstract is not fully structured. It should clearly include Methods, Results, and Conclusion sections in addition to the Background/Objective.

• The main text does not contain a separate, structured Discussion and Conclusion section. At present, the manuscript ends abruptly without a clear synthesis of findings, interpretation and limitations

• The Discussion should be organized to:

o Summarize the principal findings.

o Compare results with existing literature.

o Address strengths and limitations.

o Provide implications for practice, policy, and future research.

• The Conclusion should be concise, highlighting the key message of the study.

Without these sections, the manuscript does not meet the journal’s formatting and reporting standards. A thorough restructuring is required to ensure compliance and improve readability.

Response: We have now relabelled and otherwise reorganized the paper to conform to journal standards.

Introduction

The introduction effectively sets the context by highlighting the rising rates of perinatal substance use (PSU) and associated adverse outcomes, particularly in British Columbia. The emphasis on systemic gaps in care and the impact of stigma is well-placed.

Suggestion: Consider including a brief mention of the intersectional vulnerabilities faced by Indigenous and racialized communities in BC, which are noted later in the manuscript but could be introduced earlier to frame the urgency of the research.

Question: The authors mention that PSU is under-researched due to ethical concerns and healthcare system failures. Could they elaborate on what specific ethical concerns have historically impeded research in this area?

Response: disclosure and the potential for child removal.

• Refer to structural barriers surrounding PSU population impeding their participation in research, particularly mothers.

• Elaborate on impacts for other subgroups within this population (i.e., Indigenous people)

Funding Statement Placement

The authors currently places the funding information at the end of the Introduction, which is not consistent with the PLOS ONE guidelines. According to the journal’s requirements, funding details should be presented under a separate heading (e.g., Funding) in the appropriate section of the manuscript, not embedded within the Introduction.

For example, instead of including text such as: “Funding for this cohort was obtained from the Health Canada Substance Use and Addiction Program (2223-HQ-000028; 1819-HQ-000036).”

at the end of the Introduction, the authors should move this statement to a dedicated Funding section (usually after the Acknowledgments or before Competing Interests).

Recommendation:

• Provide a clear heading (Funding) in line with PLOS formatting.

• Place the funding statement in the correct section, ensuring consistency with other required declarations (Competing Interests, Ethics Statement, Data Availability).

• Keep the Introduction focused on background and rationale, without administrative details

This applies to the Ethical Approval section as well.

Response: Done, as requested.

Methods

Cohort Identification and Substance Use Definition:

o The authors have used a 12-month window prior to the first pregnancy-related healthcare contact to identify substance use. While this approach aims to capture ongoing use, it may also include individuals with historical or sporadic use that is not necessarily active during pregnancy.

o Suggestion: Consider discussing the potential for misclassification and how this may affect the cohort’s representativeness. Sensitivity analyses using narrower time windows (e.g., during pregnancy only) could strengthen the validity of substance use identification.

o Question: Were any validation studies conducted to assess the accuracy of the case-finding algorithm in identifying active substance use during pregnancy?

Response: Unfortunately, validation is not possible as no ‘gold standard’ measurement (ie. via self report) is feasible given the de-identified nature of the data. We have added the following to the discussion section (page 22, line 295): ““Moreover, our classifications of substance use disorders and administrative loss to follow-up may result in misclassification. We will consider alternative thresholds to assess the robustness of results to these thresholds.”

Data Linkage and Missing Information:

o The cohort links data from ten administrative databases, which is a strength. However, the absence of linkage to the BC Ministry of Child and Family Development is noted as a limitation, as it precludes identification of state-initiated child apprehensions.

o Suggestion: The authors should elaborate on how they plan to address this in future updates, and whether partnerships with child welfare agencies are being pursued.

o Question: What proportion of children in the cohort are estimated to be affected by child welfare involvement, and how might this bias the findings related to child separations?

Response: We have pursued such partnerships and determined these data are not suitable for research purposes. Somewhat unbelievably, they are not collected systematically and availability of provincial health numbers are inconsistent. To our knowledge such data has never been used for research purposes.

2. Loss to Follow-up and Censoring:

o The authors define administrative loss to follow-up as no health system contact for ≥66 months. While empirically derived, this cutoff may not fully capture individuals who have moved out of province or who disengage from healthcare due to stigma or systemic barriers.

o Suggestion: Consider discussing alternative methods to ascertain vital status or out-of-province migration, such as linkage to national databases or mortality registries.

o Question: Were there differences in demographic or clinical characteristics between those lost to follow-up and those retained? If so, how might this affect longitudinal analyses?

Response: Linkage to datasets held outside the province of BC is not currently possible; such actions could only be taken by the provincial government, as only de-identified data is provided to researchers. We have noted this as a limitation on page 22, line 295: “Moreover, our classifications of substance use disorders and administrative loss to follow-up may result in misclassification. We will consider alternative thresholds to assess the robustness of results to these thresholds.”

Measurement of Maternal-Child Separation:

o The use of geographic mismatches and social assistance records to infer separations is innovative but indirect. The authors note that they cannot distinguish between voluntary separations, kinship care, and foster care.

o Suggestion: Acknowledging this limitation is important, and future qualitative or mixed-methods studies could help contextualize these separations.

o Question: How were inconsistencies between separation indicators (e.g., SDPR vs. geographic mismatch) resolved, and what is the estimated agreement between these measures?

Response: This work is currently underway – we did not elaborate on details as we are assessing not only concordance between separation indicators, but also assessing use of geographic indicators from different component datasets within our linked data. We intend to report on this effort more exhaustively in forthcoming manuscripts, we have added the following (page 13, line 188): “Later separations can be inferred based on the geographic location of the mother and child in their individual health records (available in multiple component datasets; subject to evaluations of concordance) or by assessing whether a mother-child pair are attached to the same social assistance record.”

Overall

The methods are generally robust and well-described, and the cohort represents a significant contribution to perinatal substance use research. Addressing the above points will enhance the clarity, validity, and utility of the cohort profile for future users.

Results

• The descriptive tables are comprehensive and well-organized. The stratification by substance type is particularly useful.

• Notable Findings: The high prevalence of mental health disorders among mothers (8%) and children (26.3%), along with the substantial rates of maternal mortality (3.6%) and child separation (60%), are striking and underscore the cohort's public health relevance.

• Suggestion: In Table 1, consider presenting the proportion of mothers with polysubstance use (e.g., opioids + stimulants) to better characterize the complexity of substance use patterns.

• Question: The median follow-up time varies substantially by substance type (e.g., 6 years for cannabis-only vs. 13 years for "other substances"). What might explain these differences, and how will the analyses account for varying follow-up durations?

Response: RE: the suggestion – given the large extent of overlap and potential for many additional columns we chose to maintain the current table structure (which was in fact reduced from prior iterations in which combinations wer

---

## [Editor Report · Decision Letter 1]

15 Apr 2026

Cohort profile: mothers who use substances and their children in British Columbia, Canada

PONE-D-25-53902R1

Dear Dr. Nosyk,

We’re pleased to inform you that your manuscript has been judged scientifically suitable for publication and will be formally accepted for publication once it meets all outstanding technical requirements.

Kind regards,

David T. Zhu

Academic Editor

PLOS One
---

## [Editor Report · Acceptance letter]

PONE-D-25-53902R1

PLOS One

Dear Dr. Nosyk,

I'm pleased to inform you that your manuscript has been deemed suitable for publication in PLOS One. Congratulations! Your manuscript is now being handed over to our production team.

Kind regards,

on behalf of

Mr. David T. Zhu

Academic Editor

PLOS One